# Social support and technophobia in older patients with coronary heart disease: The mediating roles of eHealth literacy and healthcare technology self-efficacy

**Jianchun Zhao◍, Danqing Hu◍, Haowei Du, Haichao Wang, Xiaomin Tu, Aimin Wang◍** *

School of Nursing, Qingdao University, Qingdao, Shandong Province, People's Republic of China

◍ These authors contributed equally to this work.
* wam@qdu.edu.cn

## Abstract

### Objectives

The purpose of this study was to explore the relationship between social support, eHealth literacy, healthcare technology self-efficacy, and technophobia. It also analyzed the mediating effect of eHealth literacy and healthcare technology self-efficacy between social support and technophobia.

### Methods

Older patients with coronary heart diseases (n = 396) from four communities in Qingdao were interviewed using the Technophobia Scale, Social Support Rating Scale, eHealth Literacy Scale and Healthcare Technology Self-Efficacy Scale. Data were analyzed using common method deviation test, Pearson's bivariate correlation analysis, and mediation analysis using the PROCESS macro.

### Results

Social support was significantly positively correlated with eHealth literacy (r = 0.614, p < 0.01) and healthcare technology self-efficacy (r = 0.635, p < 0.01), and significantly negatively correlated with technophobia (r = −0.578, p < 0.01). eHealth literacy was significantly positively correlated with healthcare technology self-efficacy (r = 0.822, p < 0.01), and significantly negatively correlated with technophobia (r = −0.651, p < 0.01). Healthcare technology self-efficacy was significantly negatively correlated with technophobia (r = −0.700, p < 0.01). Social support had a total indirect effect on technophobia of −0.410, with eHealth literacy and healthcare technology self-efficacy mediating 24.9% and 30.2% of this effect respectively, and the chain mediating effect accounting for 44.9%.

**Data availability statement:** All relevant data are within the manuscript and its Supporting Information files.

**Funding:** AW Upper-level Project of the Natural Science Foundation of Shandong Province grant number: ZR2023MG071 URL: http://kjt.shandong.gov.cn/ the founder gave a financial support in paper submission. DH Youth Fund of the Natural Science Foundation of Shandong Province grant number: ZR2023QG027 URL: http://kjt.shandong.gov.cn/ the founder gave a financial support in paper submission. The funders had no role in study design, data collection and analysis, decision to publish, or preparation of the manuscript.

**Competing interests:** The authors have declared that no competing interests exist.

## Conclusions

Our findings provide a theoretical reference for nursing to develop appropriate interventions to alleviate technophobia among older patients with CHD.

## Introduction

Coronary heart disease (CHD) remains a leading cause of mortality worldwide, particularly among the elderly [1]. In China, the mortality rate of cardiovascular diseases accounts for the first cause of death, with a prevalence of CHD among individuals aged 60 and older reaching 27.8% [2]. To manage risk factors, minimize readmissions and mortality, and improve prognosis, older patients with CHD require self-management and care services encompassing medication, nutrition, exercise, and psychological support [3]. With the deepening of aging, the demand for healthcare services among older patients with CHD has increased, far exceeding that of other age groups. Traditional healthcare services for CHD are based on outpatient clinics, hospital wards, or rehabilitation centers, where healthcare professionals provide disease treatment, medication management, dietary guidance and health education [4]. However, older patients with CHD may be reluctant to seek medical care due to the distance from healthcare facilities or the high costs involved, which leads to a higher risk of recurrent events and hospitalizations, ultimately diminishing their chances of survival and quality of life [5]. Additionally, in China, there is an imbalance in the distribution of healthcare resources and a shortage of necessary infrastructure and specialized staff [6]. Consequently, the traditional way of accessing medical information and assistance is burdensome for healthcare organizations. To address this escalating need, innovative strategies are imperative. Digital health technology has introduced new modes of self-management for older patients with CHD, including teleconsultations, mobile applications, and wearable technologies [7].

Compared with traditional cardiac rehabilitation interventions, digital health technology interventions break through constraints of time and space and extend care services from hospitals to patients' homes [8]. Medical personnel can timely understand patients' latest health status and different needs, providing targeted health-related knowledge and symptom management to improve patients' self-management ability and quality of life [9]. A Chinese study found that patients with CHD who actively used self-management mobile applications as part of their digital health interventions had significantly higher medication adherence over 12 months compared to those who did not [10]. Most patients and healthcare professionals believe that digital health technologies can provide convenient and effective medical services [11]. However, the adoption(or anticipation of adoption) of diverse technologies can evoke negative psychological feelings, such as fear, stress, and anxiety [12]. This phenomenon is particularly pronounced among older adults, who often encounter technology later in life and, due to physical, mental, and cognitive decline, may lack confidence, thereby exacerbating these negative emotions [13–15].

In this context, technophobia emerges as a key issue. Technophobia constitutes an irrational fear and/or anxiety arising in individuals as a consequence of encountering new technologies that modify or disrupt their customary routines in executing specific tasks, manifesting either through active physical reactions akin to avoidance or passive emotional states like distress or apprehension [12]. Elderly patients with heightened technophobia, who may initially be afraid of making mistakes when operating the technology and experiencing negative consequences, have their fear of using health technology further exacerbated upon receiving negative feedback, ultimately leading to avoidance behaviors [16]. Research shows technophobia reduces the perceived ease of technology use and willingness to adopt digital health technology [17]. Given its potential to hinder older patients with CHD from benefiting from these technologies, identifying the protective factors against technophobia to reduce its impact on this population is essential. As an important coping resource, social support is considered a key external protective factor [18,19].

Social support has been found to influence technology adoption and utilization among individuals, and there is a significant negative correlation between social support and technophobia in older adults [20]. A mixed-methods study found that a major cause of technophobia among older adults was the absence of guidance on using technology, whereas technological guidance and emotional support provided by younger people increased eHealth literacy and self-efficacy, and reduced levels of technophobia among older adults [21]. Therefore, older adults with more social support have wider access to new technology resources and information, which to some extent has a positive impact on their acceptance of digital health technologies [22].

Apart from social support, eHealth literacy is also an essential factor influencing technophobia. eHealth literacy refers to the ability to use digital media to search, evaluate health information, and make informed health decisions [23]. A study reveals a direct link between eHealth literacy and technophobia [20]. Individuals with higher levels of eHealth literacy usually have some experience using technology and can access the health information they want through simple operation. By adopting health information, patients can increase their awareness of disease and ultimately improve their health behaviors [24]. Additionally, healthcare technology self-efficacy is a key internal protective factor affecting technophobia, playing a critical role in guiding technology use behavior [25]. Defined as an individual's confidence in using digital healthcare technology, healthcare technology self-efficacy provides a more sensitive measure of an individual's confidence in using digital healthcare technology within a healthcare setting compared to general self-efficacy [26]. A previous study found a negative correlation between self-efficacy and technophobia in older adults, with those high in self-efficacy exhibiting lower levels of technophobia [27]. Furthermore, eHealth literacy is a protective factor for the self-efficacy of older adults in using health technology [26]. Self-efficacy theory states that the most critical factor influencing self-efficacy is prior experience [28]. Older adults with higher eHealth literacy increase their self-efficacy by receiving positive feedback on their use of technology, resulting in greater confidence in accepting and using new technology [29].

Previous studies have shown that social support, eHealth literacy, and healthcare technology self-efficacy play an important role in influencing technophobia in older patients with CHD. The relationship between social support, eHealth literacy, and self-efficacy has also been established, particularly in the self-management of older patients with chronic diseases and their use of mobile health technologies [19,30]. However, to our knowledge, the underlying relationship between these factors and technophobia remains unexplored in existing literature. According to the social ecosystem theory [31], the external environment can influence individual cognition and behavior, which in turn affects the psychological state of older adults when facing technology [27]. Consequently, this study investigated the mediating effect of eHealth literacy and healthcare technology self-efficacy between social support and technophobia to offer theoretical and empirical evidence for mitigating technophobia in older patients with CHD through interventions targeting protective factors. In light of the research evidence, we propose four hypotheses: (H1) social support is related to technophobia; (H2) eHealth literacy may play a mediating role between social support and technophobia; (H3) healthcare technology self-efficacy may mediate between social support and technophobia; (H4) eHealth literacy and healthcare technology self-efficacy may have a chain mediating effect between social support and technophobia.

## Materials and methods

### Participants

We conducted a cross-sectional analysis of self-report data collected from December 2023 to May 2024. Older patients with CHD who met the inclusion criteria in four communities in Qingdao City were selected as the participants of this study using convenience sampling. Qingdao is an economically developed city in northern China with a population of about 10 million, of which the elderly account for 23.8% of the total population [32]. The four communities were randomly selected from each of Qingdao's four main municipal districts (Shinan, Shibei, Laoshan, Licang), and potential participants lived in both urban and rural communities. The inclusion criteria were participants that: (a) aged ≥60 years; (b) meet the diagnostic criteria in the Clinical Guidelines for the Diagnosis and Treatment of CHD; (c) are conscious and can communicate through words or language; and (d) informed consent and willingness to cooperate with the study. Exclusion criteria included: (a) in the acute attack period of CHD; (b) with the combination of other systematic serious diseases; and (c) had visual and auditory disorders or mental impairment.

### Measures

**Demographic characteristics.** Based on a comprehensive review of the literature, we selected demographic factors that may influence the outcome variables and independently developed a questionnaire. The questionnaire included sociodemographic items that collected relevant information about participants' characteristics, including gender, age, marital status, residence, education level, work status, family structure, and family monthly income.

**Technophobia.** The Technophobia Scale was used to assess technophobia [33]. The Chinese version of the Technophobia Scale contains 13 items and 3 dimensions: techno-anxiety, techno-paranoia, and privacy concerns, with responses on a five-point Likert scale from 1 ("strongly disagree") to 5 ("strongly agree"). The total score ranges from 13 to 65 points, with higher scores representing higher levels of technophobia. The Cronbach's α coefficient was 0.91 for the total scale and 0.88, 0.83, and 0.75 for the three factors, respectively. These values are greater than the acceptable value of 0.70, which indicates that the Chinese version of the Technophobia Scale has good reliability. The Cronbach's α of the Technophobia Scale in this study was 0.89.

**Social support.** The Social Support Rating Scale (SSRS) which was compiled by Xiao in 1994 [34], was used to measure social support. The scale includes 10 items and 3 dimensions: subjective support, objective support, and utilization of support. Items 1–4 & 8–10: Select one option per item (1–4 points). Item 5: A-D options (4-point scale: 1 = none to 4 = full support). Items 6–7: 0 points without sources; score = number of sources listed. The total score on the scale ranges from 12 to 66, with higher scores representing more social support. A total score of 12–22 indicates a low level of social support, 23–44 indicates a medium level of social support, and 45–66 indicates a high level of social support. The SSRS has good reliability and validity with Cronbach's α of 0.89 to 0.94. The Cronbach's α of this scale in this study was 0.87.

**eHealth literacy.** The eHealth Literacy Scale (eHEALS) was used to evaluate eHealth literacy [35]. The Chinese version of eHEALS includes 8 items and 3 dimensions: application ability, evaluation ability, and decision-making ability. Each item is scored on a five-point Likert scale from 1 ("strongly disagree") to 5 ("strongly agree"). The total score ranges from 8 to 40 points, with higher scores indicating a higher level of eHealth literacy. Cronbach's α for the Chinese version of eHEALS was 0.91, and the Cronbach's α for the eHEALS in this study was 0.98.

**Healthcare technology self-efficacy.** The Healthcare Technology Self-Efficacy Scale was used to assess healthcare technology self-efficacy [26]. Our group has revised the Chinese version, which can be used to assess the confidence of individuals when using health technology or receiving services provided by health technology. The scale contains 12 items and 3 dimensions: technology, service, and web. A 5-point Likert scale was used, ranging from 1 = "Strongly Disagree" to 5 = "Strongly Agree", with entries 3, 6, and 10 reverse-scored. The total score ranges from 12 to 60 points, with higher

scores indicating higher self-efficacy in health technology. The Cronbach's alpha coefficient of the Chinese version of the Healthcare Technology Self-Efficacy Scale was 0.93, the split-half reliability was 0.81, and the re-test reliability after two weeks was 0.89. The Cronbach's α of this scale was 0.89 in this study.

## Data collection

The study was approved by the Ethics Committee of Qingdao University School of Nursing (QDU-HEC-2023245). Data collection was done through face-to-face distribution of paper questionnaires. Participants were recruited from the community health service centre after approval by the community manager. The physician reviewed the participants' health records and interviewed the patients to determine if the criteria were met. The researcher then explained the purpose of the study to the patients who met the criteria and obtained their consent. All participants were informed that it was an anonymous survey and they had the right to refuse to participate or withdraw at any time during the study. During the data collection process, researchers instructed participants to fill out paper questionnaires. For those who were unable to fill out the questionnaire on their own, the researcher read the survey items to them and recorded their answers by unified instruction. The researcher checked the completed questionnaires immediately and asked the participants to provide any missing data. Questionnaires with apparent regularities and logical errors were eliminated, such as a questionnaire with at least a string of more than 10 consecutive identical item responses. All paper questionnaires with the informed consent form were only accessible to the research team to ensure security and confidentiality. A total of 412 questionnaires were distributed and 396 were validly collected, with a valid recovery rate of 96.1%.

## Statistical analyses

SPSS 25.0 and AMOS were used for data analysis. In order to improve the rigor of the study, we tested for common method bias first. Common method bias refers to artifactual covariation between a predictor and a valid scale variable because of the same data source or rater, the same measurement environment, the context of the item, and the characteristics of the item itself. Because this study collected data through self-reporting methods, it was possible that there could be an issue with common method bias. Podsakoff et al. recommend introducing a method factor for testing [36]. Therefore, we built a two-factor model in AMOS by adding a method factor as a global factor to the originally designed factors. If the model fit indices become much better after the addition of the method factor (e.g., CFI and TLI increase by more than 0.1 and RMSEA decrease by more than 0.05), it indicates that there is a serious common method bias [37].

Continuous variables are represented by (mean ± standard deviation), and categorical variables are represented by frequency and percentages. Independent samples t-test or one-way ANOVA were used to compare differences in the demographic characteristics. In the mediation effects analysis that followed, variables that showed significant differences in demographic characteristics were controlled for as covariates.

To verify the research hypotheses, the analyses were conducted in two stages. Firstly, Pearson's bivariate correlation analysis was used to analyze the relationship between variables. Secondly, the chain mediation effect was tested using Model 6 from the SPSS-PROCESS macro program, which refers to the indirect effect in a causal pathway where the influence of an independent variable (X) on a dependent variable (Y) is transmitted sequentially through multiple mediators (e.g., $M_1$, $M_2$) in a specified order. The PROCESS macro employs a stepwise approach for mediation testing, followed by a Bootstrap 95% confidence interval (CI) to determine the indirect effects based on 5000 bootstrapped samples. As a non-parametric resampling procedure, bootstrapping is considered the most powerful method for small samples because it is the least vulnerable to type 1 errors. The 95% CI is a statistical range used to estimate the plausible values of mediation effect. If the study were repeated 100 times, approximately 95 of the calculated intervals would contain the true parameter value. If the 95% CI did not include 0, the effects were considered significant ($P < 0.05$). The SPSS-PROCESS macro, developed by Andrew F. Hayes in 2013, has been widely validated in the literature for its robust analytical capabilities [38]. This program covers nearly a hundred mediation and moderation models and simplifies the analysis process,

providing comprehensive results quickly. Its main advantage lies in the ability to perform bias-corrected, non-parametric percentile bootstrap tests and to provide the specific path coefficients for each mediator, along with the individual mediation effect test results for each mediator variable.

## Results

### Common method deviation test

We established a two-factor model, and model M2 was constructed by adding method factors to the original validated factor analysis model M1. Then comparing the model fit indices of models M1 and M2: ΔCFI = 0, ΔTLI = 0.016, neither exceeding 0.1; ΔRMSEA = 0.003, not exceeding 0.05. It indicated that the model did not improve significantly after adding the method factors and there was no serious common method deviation in the data used in this study.

### Technophobia of participants with different demographic characteristics

A total of 396 older patients with CHD were included in this study. Table 1 shows the characteristics of the participants as well as the mean, SD, and univariate analysis of technophobia. Participants' average age was 69.78 (SD = 6.36) years old (range 60–88 years). Three-quarters of patients lived in the city. Most patients were married and cared for their spouses and children. Notably, differences in technophobia scores were significant among older patients with CHD by age, residence, work status, educational level, and family monthly income ($p < 0.05$).

### Correlation analysis of technophobia, social support, eHealth literacy and healthcare technology self-efficacy

Table 2 shows the results of the Pearson correlation analysis between social support, eHealth literacy, healthcare technology self-efficacy and technophobia. Social support was significantly positively correlated with eHealth literacy ($r = 0.614$, $p < 0.01$) and healthcare technology self-efficacy ($r = 0.635$, $p < 0.01$), and significantly negatively correlated with technophobia ($r = -0.578$, $p < 0.01$). eHealth literacy was significantly positively correlated with healthcare technology self-efficacy ($r = 0.822$, $p < 0.01$), and significantly negatively correlated with technophobia ($r = -0.651$, $p < 0.01$). Finally, healthcare technology self-efficacy was significantly negatively correlated with technophobia ($r = -0.700$, $p < 0.01$).

### Test of the mediating effect of eHealth literacy and healthcare technology self-efficacy

Data were analyzed using model 6 in the SPSS plug-in PROCESS (version 4.1), with social support as the independent variable, technophobia as the dependent variable, eHealth literacy and healthcare technology self-efficacy as the chained mediator variables, and statistically significant variables from the univariate analysis as control variables. The chain mediation model of eHealth literacy and healthcare technology self-efficacy between social support and technophobia is shown in Fig 1. Before adding the mediator variable, social support negatively influenced technophobia ($β = -0.441$, $p < 0.001$). The regression results (Table 3) showed that the social support for older patients with CHD had a significant direct predictive effect on technophobia ($β = -0.170$, $p < 0.001$). The social support positively predicted eHealth literacy ($β = -0.445$, $p < 0.001$) and healthcare technology self-efficacy ($β = -0.198$, $p < 0.001$). eHealth literacy positively predicted healthcare technology self-efficacy ($β = -0.661$, $p < 0.001$) and negatively predicted technophobia ($β = -0.152$, $p < 0.05$). Finally, healthcare technology self-efficacy negatively predicted technophobia ($β = -0.413$, $p < 0.001$).

The results of mediating effects analysis using the Bootstrap method of bias correction are shown in Table 4. The direct effect of social support on technophobia was −0.257 and the direct effect was significant. eHealth literacy and healthcare technology self-efficacy played a partial mediating role, with a total indirect effect value of −0.410. The three paths of the mediating effect were specified as follows: ① social support → eHealth literacy → technophobia (effect value = −0.102, 95% CI [−0.199 to −0.007]), meant that this mediating effect was statistically significant; ② social support → healthcare technology self-efficacy → technophobia (effect value = −0.124, 95% CI [−0.191 to −0.065]), and the mediating effect of healthcare

**Table 1. Technophobia of participants with different demographic characteristics.**

| Variables | | N (%) | Technophobia ($\bar{x} \pm s$) | t/F | p |
|---|---|---|---|---|---|
| **Age** | | | | −4.271 | <0.001 |
| | 60~69 | 202 (51.0) | 35.58±11.69 | | |
| | 70~ | 194 (49.0) | 40.58±11.61 | | |
| **Gender** | | | | −1.072 | 0.284 |
| | Male | 196 (49.5) | 37.38±11.98 | | |
| | Female | 200 (50.5) | 38.67±11.82 | | |
| **Residence** | | | | 5.416 | <0.001 |
| | Rural | 93 (23.5) | 43.30±10.32 | | |
| | City | 303 (76.5) | 36.41±11.90 | | |
| **Marital status** | | | | −1.350 | 0.178 |
| | Married | 351 (88.6) | 37.74±11.89 | | |
| | single | 45 (11.4) | 40.28±11.83 | | |
| **Type of residence** | | | | 1.277 | 0.282 |
| | Solitary | 29 (7.3) | 38.34±11.58 | | |
| | Residence with spouse only | 307 (77.5) | 37.88±11.82 | | |
| | Residence with children only | 18 (4.5) | 43.11±12.02 | | |
| | Residence with spouse and children | 42 (10.6) | 36.76±12.50 | | |
| **Work status** | | | | 34.669 | <0.001 |
| | Mental labor | 104 (26.3) | 32.68±10.82 | | |
| | Physical labor | 169 (42.7) | 43.23±10.59 | | |
| | Partly mental and partly physical labor | 123 (31.1) | 35.41±11.68 | | |
| **Educational level** | | | | 29.939 | <0.001 |
| | Elementary school and below | 103 (26.0) | 45.00±10.04 | | |
| | Junior high schools | 139 (35.1) | 40.19±11.01 | | |
| | High school/technical secondary school | 72 (18.2) | 33.75±10.50 | | |
| | Junior college | 48 (12.1) | 29.27±10.58 | | |
| | Bachelor degree or above | 34 (8.6) | 29.52±9.08 | | |
| **Family monthly income(RMB)[a]** | | | | 40.912 | <0.001 |
| | 0~6000 | 78 (19.7) | 44.16±10.33 | | |
| | 6000~10000 | 181 (45.7) | 40.30±11.12 | | |
| | 10000~ | 137 (34.6) | 31.54±10.79 | | |

Note: $\bar{x} \pm s$: mean±standard deviations.

[a]6,000 RMB is approximately 846 US dollars, 10,000 RMB is approximately 1410 US dollars.

technology self-efficacy was significant; ③ social support→eHealth literacy→healthcare technology self-efficacy→technophobia (effect value=−0.184, 95% CI [−0.267 to −0.113]), indicated that eHealth literacy and healthcare technology self-efficacy had a significant chain-mediated effect in the influence of social support on technophobia in older patients with CHD.

## Discussion

The results of this study showed that the technophobia score among older patients with CHD was 38.03±11.90, which was at a moderate level, consistent with that reported by Peng et al. [27]. Before introducing the mediator variable, the

**Table 2. Statistical description and related analysis results.**

| Variables | x̄±s | 1 | 2 | 3 | 4 |
|---|---|---|---|---|---|
| 1. Technophobia | 38.03±11.90 | 1.000 | | | |
| 2. Social support | 36.40±7.87 | −0.578** | 1.000 | | |
| 3. eHealth literacy | 18.18±10.11 | −0.651** | 0.614** | 1.000 | |
| 4. Healthcare technology self-efficacy | 33.08±11.39 | −0.700** | 0.635** | 0.822** | 1.000 |

Note: x̄±s: mean±standard deviations.

**p<0.01.

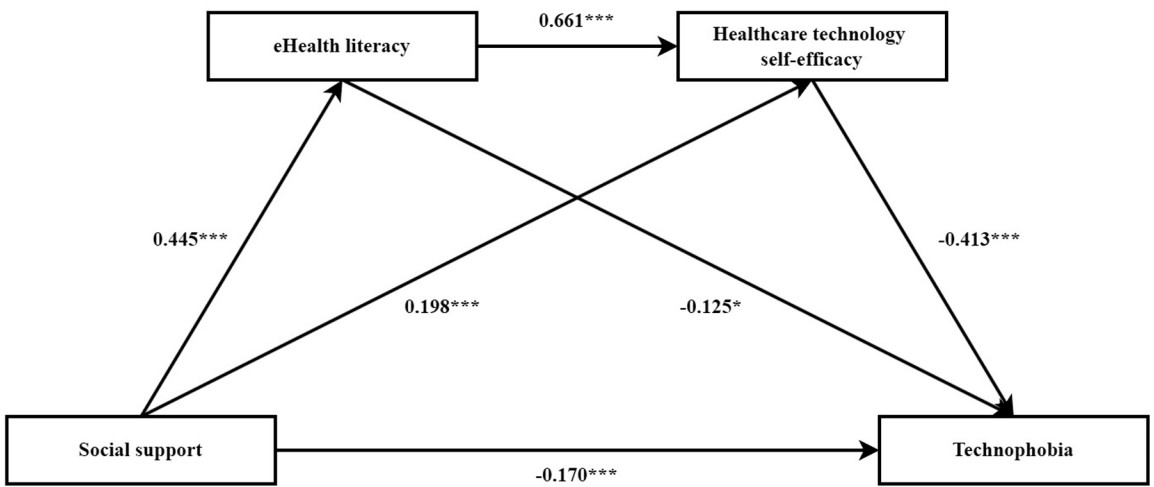

**Fig 1. The chain mediation model of eHealth literacy and healthcare technology self-efficacy between social support and technophobia.** Note: All the coefficients are standerdized. *$p<0.05$, ***$p<0.001$.

effect value of social support on technophobia was −0.441. Additionally, after adding the mediator variable, the direct effect accounted for 38.5% (−0.257) of the total effect, even without considering the mediator variable, social support can significantly reduce technophobia. The result was consistent with previous research, which observed reduced technophobia in elderly cancer patients with higher levels of family support and social engagement [39]. These findings supported hypothesis 1. Due to cognitive and physical decline, older patients with CHD may experience psychological stress when confronted with emerging technologies [40]. According to the buffer model in social support theory, social support can mitigate the impact of stressful events on individuals by either reducing the cognitive appraisal of stress or providing solutions to address specific issues [41]. Kwan et al. underscored the mental health benefits of social support and its role in preventing anxiety symptoms [22]. Older patients with CHD with robust social support networks can draw upon emotional and practical assistance when learning digital health technologies, thereby reducing fear of technologies, as evidenced by Lee et al. [21].

Mediation effect analysis found that eHealth literacy partially mediated between social support and technophobia, with its individual mediation effect accounting for 24.9% of the total indirect effect (validating hypothesis 2). This indicates that social support can directly influence technophobia in older patients with CHD while also exerting an indirect effect through eHealth literacy. This may be due to the fact that older patients with greater social support are more likely to have better access to health information and resources, thereby avoiding the negative emotions caused by low eHealth literacy. When older patients with CHD feel supported by family, friends, and society, they are better equipped to face the challenges of

**Table 3. Regression analysis among variables in the chain intermediary model.**

| Regression equation | | Model fit indices | | | Regression coefficient | |
|---|---|---|---|---|---|---|
| **Outcome variable** | **Predictor variable** | **R** | **R²** | **F** | **β** | **t** |
| **eHealth literacy** | | 0.697 | 0.486 | 61.217 | | |
| | Social support | | | | 0.445 | 10.289*** |
| | Age | | | | −0.059 | −1.491 |
| | Residence | | | | −0.099 | −2.400* |
| | Work status | | | | −0.046 | −1.226 |
| | Education level | | | | 0.290 | 5.834*** |
| | Family monthly income | | | | 0.002 | −0.035 |
| **Healthcare technology self-efficacy** | | 0.841 | 0.708 | 134.202 | | |
| | Social support | | | | 0.198 | 5.387*** |
| | eHealth literacy | | | | 0.661 | 17.280*** |
| | Age | | | | −0.022 | −0.734 |
| | Residence | | | | −0.047 | −1.485 |
| | Work status | | | | 0.010 | 0.362 |
| | Education level | | | | 0.043 | 1.097 |
| | Family monthly income | | | | −0.004 | −0.121 |
| **Technophobia** | | 0.738 | 0.545 | 65.848 | | |
| | Social support | | | | −0.170 | −3.569*** |
| | eHealth literacy | | | | −0.152 | −2.382* |
| | Healthcare technology self-efficacy | | | | −0.413 | −6.518*** |
| | Age | | | | 0.019 | −0.496 |
| | Residence | | | | 0.010 | 0.245 |
| | Work status | | | | −0.073 | −2.045* |
| | Education level | | | | −0.064 | −1.315 |
| | Family monthly income | | | | −0.085 | −1.915 |

Note: All the coefficients are standerdized.

*$p < 0.05$,

**$p < 0.01$,

***$p < 0.001$. Social support as the independent variable. Technophobia as the dependent variable. eHealth literacy and healthcare technology self-efficacy as the chained mediator variables. Age, Residence, Work status, Education level and Family monthly income as control variables.

**Table 4. The mediating effect of eHealth literacy and healthcare technology self-efficacy between social support and technophobia.**

| | Effect size | SE | 95%CI | Relative mediation effect % |
|---|---|---|---|---|
| **Total effect** | −0.668[a] | 0.071 | −0.806 ∼ −0.529 | 100.0% |
| **Direct effect** | −0.257[a] | 0.072 | −0.399 ∼ −0.116 | 38.5% |
| **Total mediation effect** | −0.410[a] | 0.056 | −0.524 ∼ −0.304 | 61.5% |
| **Indirect effect 1** | −0.102[a] | 0.049 | −0.199 ∼ −0.007 | 24.9% |
| **Indirect effect 2** | −0.124[a] | 0.032 | −0.191 ∼ −0.065 | 30.2% |
| **Indirect effect 3** | −0.184[a] | 0.039 | −0.267 ∼ −0.113 | 44.9% |

Note: SE: Standard Error; CI: confidence interval.

[a]An empirical 95% confidence interval does not overlap with zero.

Indirect effect 1: social support → eHealth literacy → technophobia.

Indirect effect 2: social support → healthcare technology self-efficacy → technophobia.

Indirect effect 3: social support → eHealth literacy → healthcare technology self-efficacy → technophobia.

new technology [42]. Studies have shown that health knowledge seeking and emotional support can all improve eHealth literacy [43], and a high level of eHealth literacy can help reduce technophobia [44].

Healthcare technology self-efficacy also played a partial mediating role between social support and technophobia, with its individual mediating effect accounting for 30.2% of the total indirect effect (validating hypothesis 3). The result was consistent with previous research, which found that individuals with higher healthcare technology self-efficacy have greater confidence in mastering the utilization of health technology, subsequently making them less prone to experiencing fear and avoidance [25]. Self-efficacy theory suggests that self-efficacy not only has an impact on individuals' behaviors and decision-making but also has a direct effect on their psychological responses during activities [28]. Encouragement from family and friends can enhance the self-efficacy of older patients with CHD, enabling them to face challenges with greater confidence and resilience, thereby reducing the occurrence of technophobia [27].

This research found that eHealth literacy and healthcare technology self-efficacy jointly played a chain mediating role in the influence of social support on technophobia among older patients with CHD, with the mediating effect accounting for 44.9% of the total indirect effect (supporting hypothesis 4). This suggests that social support influencing technophobia through the chain-mediated effects of eHealth literacy and healthcare technology self-efficacy is the predominant indirect pathway, contributing nearly half of the total indirect effect. Sequential pathways can explain this mediating effect. Specifically, social support provides older patients with CHD with the confidence to face difficulties and challenges, effectively alleviating psychological stress related to their illness [45]. This gives patients the confidence to recover and a willingness to access digital technologies such as the Internet. Additionally, social support can offer guidance and assistance in using technology, which helps to improve eHealth literacy [30]. Enhanced eHealth literacy leads to positive experiences with technology, strengthening patients' self-efficacy in using medical technology [46]. As a result, patients are more likely to use new technologies with confidence, reducing the likelihood of negative emotions or avoidance behaviors [27].

According to the findings, to alleviate technophobia among older patients with CHD, the relevant departments of hospitals and communities should establish a comprehensive social support system for them. Healthcare professionals should encourage intergenerational interaction between patients and family members, especially with younger generations, to facilitate digital technology communication [47]. Peer-based technology support groups should be established for older patients with CHD, facilitating experience sharing through both offline activities and online communities [48]. Community healthcare centers can organize training sessions on technological skills, providing patients with spaces for learning and interaction [48]. In addition, the findings suggest that the chain-mediated effect of eHealth literacy and healthcare self-efficacy is the core mechanism. When reducing technophobia through social support, efforts should focus on these two aspects. Family members and professionals should prioritize improving patients' basic technical competence and critical thinking skills to identify misinformation [21]. Instructors should provide immediate encouragement after patients master each technical skill to enhance self-efficacy. For anxiety management, instructors can guide patients to adopt positive self-affirmations during technology use.

## Study limitations

The current study also has some limitations. First, the cross-sectional study design limited the inference of causal relationships between variables. Future studies should adopt an intervention or longitudinal approach to examine the real causal relationship. Second, the convenient sampling method may have introduced selection bias. Future studies should consider probability sampling methods such as simple random sampling and stratified sampling. Furthermore, although the participants came from different communities, they were recruited from a single province in China. Because older patients with CHD in community outpatient clinics may have milder conditions compared to those in hospital wards, and we excluded individuals in the acute attack period of CHD or with other serious illnesses, this may have led to subtle differences in the results. This limits the generalizability of the results to a broader population of older patient with CHD. Future studies should include participants from diverse regions and backgrounds through multi-centre, large-scale study

designs to enhance the generalizability of findings. Finally, most of the data were collected through researcher inquiry due to the low level of education and poor vision of older people, which may introduce response bias. Future studies should use more objective measures combined with qualitative, observational and experimental approaches to explore the interactions between social support, eHealth literacy, healthcare technology self-efficacy and technophobia.

## Conclusions

This study investigated the relationship between social support, eHealth literacy, healthcare technology self-efficacy, and technophobia. Social support affects technophobia both directly and indirectly through the mediating roles of eHealth literacy and healthcare technology self-efficacy. In this way, it provides a theoretical reference for nursing to develop appropriate interventions to alleviate technophobia among older patients with CHD.

## Supporting information

**S1 File. The dataset used in the manuscript.**
(XLSX)

## Acknowledgments

The authors would like to express their gratitude to the staff of Qingdao Community Health Centre. The authors would like to express their sincere gratitude to the older patients with CHD who volunteered to participate in this study.

## Author contributions

**Conceptualization:** Jianchun Zhao, Danqing Hu, Aimin Wang.

**Data curation:** Jianchun Zhao, Haowei Du, Haichao Wang, Xiaomin Tu.

**Formal analysis:** Jianchun Zhao, Danqing Hu, Aimin Wang.

**Funding acquisition:** Danqing Hu, Aimin Wang.

**Investigation:** Jianchun Zhao, Haowei Du, Haichao Wang, Xiaomin Tu.

**Methodology:** Jianchun Zhao, Danqing Hu.

**Supervision:** Aimin Wang.

**Writing – original draft:** Jianchun Zhao, Danqing Hu.

**Writing – review & editing:** Jianchun Zhao, Danqing Hu, Haowei Du, Haichao Wang, Xiaomin Tu, Aimin Wang.

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
