## [Decision Letter · Decision Letter 0]

1 Nov 2024

PONE-D-24-40957Social support and technophobia in older patients with coronary heart disease: The mediating roles of eHealth literacy and healthcare technology self-efficacyPLOS ONE

Dear Dr. Wang,

Thank you for submitting your manuscript to PLOS ONE. After careful consideration, we feel that it has merit but does not fully meet PLOS ONE’s publication criteria as it currently stands. Therefore, we invite you to submit a revised version of the manuscript that addresses the points raised during the review process.

We look forward to receiving your revised manuscript.

Kind regards,

Saeideh Valizadeh-Haghi, Ph.D.

Academic Editor

PLOS ONE

Journal requirements: When submitting your revision, we need you to address these additional requirements. 1. Please ensure that your manuscript meets PLOS ONE's style requirements, including those for file naming. The PLOS ONE style templates can be found at https://journals.plos.org/plosone/s/file?id=wjVg/PLOSOne_formatting_sample_main_body.pdf and https://journals.plos.org/plosone/s/file?id=ba62/PLOSOne_formatting_sample_title_authors_affiliations.pdf 2. Please include a caption for figure 1. 3. Please include a caption for table 4. 4. Thank you for stating the following financial disclosure:  [AWUpper-level Project of the Natural Science Foundation of Shandong Province grant number: ZR2023MG071URL: http://kjt.shandong.gov.cn/the founder gave a financial support in paper submission.DHYouth Fund of the Natural Science Foundation of Shandong Provincegrant number: ZR2023QG027URL: http://kjt.shandong.gov.cn/the founder gave a financial support in paper submission.].  Please state what role the funders took in the study.  If the funders had no role, please state: ""The funders had no role in study design, data collection and analysis, decision to publish, or preparation of the manuscript."" If this statement is not correct you must amend it as needed. Please include this amended Role of Funder statement in your cover letter; we will change the online submission form on your behalf. 5. Please include a copy of Table 4 which you refer to in your text on page 11.

Additional Editor Comments:

Reviewer(s) comments:

Dear authors

The submission is well-written, well-organized, and features an interesting topic. Here are some suggestions to enhance clarity and readability

Introduction

The Introduction is comprehensive and well-structured, providing a clear overview of the topic and the rationale for the study. Here are some suggestions to enhance clarity and readability:

- The opening sentence is crucial in the introduction; the researchers should consider starting with a more engaging sentence to capture the reader’s attention. For example: “coronary heart disease (CHD) remains a leading cause of morbidity and mortality worldwide, particularly among the elderly.”

- The submission needs minor revision for smoother transitions between paragraphs to ensure a cohesive flow of ideas.

- It is recommended to clarify the existing gap; researchers should succinctly outline the traditional way of accessing medical information and assistance and explain why them are burdensome for healthcare organizations. I mean before or right after this sentence:

“Consequently, the traditional way of accessing medical information and assistance burdensome for healthcare organizations.”

Material and methods

The “Materials and Methods” section is quite comprehensive and well-structured. Here are a few suggestions to improve clarity and readability:

- The researcher mention that a convenience sampling method was used. It may be useful to briefly discuss the potential limitations of this method, such as selection bias, and how it may affect the generalizability of the findings.

- The inclusion and exclusion criteria are clearly stated. However, it might be beneficial to justify the exclusion of individuals with acute CHD or other serious diseases, as this could be a significant factor affecting the nuances of your findings.

- The list of demographic characteristics is comprehensive. Consider providing a rationale for the choice of these particular variables and how they relate to your study's aims.

- The inclusion of Cronbach's alpha values is helpful. It would be useful to provide a short explanation on how these statistics indicate the reliability of the scales used. For example, a Cronbach’s alpha above 0.7 is generally considered acceptable in social science research.

- Include more details about the development or validation processes of Technophobia and Social Support Measures in the Chinese context if available. This would strengthen the credibility of the scales.

- It is good to see ethical approval included. The researcher might consider addressing any specific measures taken to ensure the confidentiality and anonymity of participants.

- The process of recruitment and data collection is well described. However, detailing how participants were informed about their rights (e.g., right to withdraw) could strengthen the ethical considerations.

- The researcher mentioned checking completed questionnaires for deficiencies or errors. Elaborating on what specific types of deficiencies or errors were checked could add thoroughness to this section.

Statistical analyses

- Mentioning the software used is good practice. The researchers may consider briefly explaining why they chose SPSS and AMOS and their appropriateness for their analyses.

- The mention of testing for common method bias is excellent. The authors might elaborate on the techniques or specific tests performed in AMOS for clarity.

- When discussing the use of the PROCESS macro for mediation analysis, consider including a brief explanation of the significance of mediation analysis, particularly for readers who may not be familiar with it.

Results

- The results section effectively addresses key points, but it is advisable to include information about each table below it.

Discussion

- The discussion is well-structured, but some sentences could be more concise for better readability.

- The researchers should be ensuring consistent use of terms. For instance, “elderly patients with CHD” and “older adults” are used interchangeably. It might be helpful to stick to one term throughout the discussion.

- The explanation of the mediating roles of eHealth literacy and healthcare technology self-efficacy is good. However, the researcher might want to elaborate on how these mediators specifically influence the relationship between social support and technophobia. For instance, they can provide more examples or scenarios illustrating these mediating effects.

- It is recommended to highlight the practical implications of the findings.

Reviewers' comments:

Reviewer's Responses to Questions

**Comments to the Author**

1. Is the manuscript technically sound, and do the data support the conclusions?

Reviewer #1: Yes

2. Has the statistical analysis been performed appropriately and rigorously? 

Reviewer #1: Yes

3. Have the authors made all data underlying the findings in their manuscript fully available?

Reviewer #1: Yes

4. Is the manuscript presented in an intelligible fashion and written in standard English?

Reviewer #1: Yes

5. Review Comments to the Author

Reviewer #1: Dear authors

The submission is well-written, well-organized, and features an interesting topic. Here are some suggestions to enhance clarity and readability

Introduction

The Introduction is comprehensive and well-structured, providing a clear overview of the topic and the rationale for the study. Here are some suggestions to enhance clarity and readability:

- The opening sentence is crucial in the introduction; the researchers should consider starting with a more engaging sentence to capture the reader’s attention. For example: “coronary heart disease (CHD) remains a leading cause of morbidity and mortality worldwide, particularly among the elderly.”

- The submission needs minor revision for smoother transitions between paragraphs to ensure a cohesive flow of ideas.

- It is recommended to clarify the existing gap; researchers should succinctly outline the traditional way of accessing medical information and assistance and explain why them are burdensome for healthcare organizations. I mean before or right after this sentence:

“Consequently, the traditional way of accessing medical information and assistance burdensome for healthcare organizations.”

Material and methods

The “Materials and Methods” section is quite comprehensive and well-structured. Here are a few suggestions to improve clarity and readability:

- The researcher mention that a convenience sampling method was used. It may be useful to briefly discuss the potential limitations of this method, such as selection bias, and how it may affect the generalizability of the findings.

- The inclusion and exclusion criteria are clearly stated. However, it might be beneficial to justify the exclusion of individuals with acute CHD or other serious diseases, as this could be a significant factor affecting the nuances of your findings.

- The list of demographic characteristics is comprehensive. Consider providing a rationale for the choice of these particular variables and how they relate to your study's aims.

- The inclusion of Cronbach's alpha values is helpful. It would be useful to provide a short explanation on how these statistics indicate the reliability of the scales used. For example, a Cronbach’s alpha above 0.7 is generally considered acceptable in social science research.

- Include more details about the development or validation processes of Technophobia and Social Support Measures in the Chinese context if available. This would strengthen the credibility of the scales.

- It is good to see ethical approval included. The researcher might consider addressing any specific measures taken to ensure the confidentiality and anonymity of participants.

- The process of recruitment and data collection is well described. However, detailing how participants were informed about their rights (e.g., right to withdraw) could strengthen the ethical considerations.

- The researcher mentioned checking completed questionnaires for deficiencies or errors. Elaborating on what specific types of deficiencies or errors were checked could add thoroughness to this section.

Statistical analyses

- Mentioning the software used is good practice. The researchers may consider briefly explaining why they chose SPSS and AMOS and their appropriateness for their analyses.

- The mention of testing for common method bias is excellent. The authors might elaborate on the techniques or specific tests performed in AMOS for clarity.

- When discussing the use of the PROCESS macro for mediation analysis, consider including a brief explanation of the significance of mediation analysis, particularly for readers who may not be familiar with it.

Results

- The results section effectively addresses key points, but it is advisable to include information about each table below it.

Discussion

- The discussion is well-structured, but some sentences could be more concise for better readability.

- The researchers should be ensuring consistent use of terms. For instance, “elderly patients with CHD” and “older adults” are used interchangeably. It might be helpful to stick to one term throughout the discussion.

- The explanation of the mediating roles of eHealth literacy and healthcare technology self-efficacy is good. However, the researcher might want to elaborate on how these mediators specifically influence the relationship between social support and technophobia. For instance, they can provide more examples or scenarios illustrating these mediating effects.

- It is recommended to highlight the practical implications of the findings.

6. PLOS authors have the option to publish the peer review history of their article (what does this mean? ). If published, this will include your full peer review and any attached files.

**Do you want your identity to be public for this peer review?** For information about this choice, including consent withdrawal, please see our Privacy Policy .

Reviewer #1: No

---

## [Author Response · Author response to Decision Letter 1]

27 Nov 2024

Response: Thank you for your suggestions, we have scrutinized the formatting of the manuscript and identified a few minor issues which have been corrected in the revised manuscript. Specific amendments are as follows: add a symbol legend for the author's signature “1” [Page1, Line 9,13]; Change the first level heading “Abstract” “Study limitations” to font size 18. [Page18, Line 413]

2. Please include a caption for figure 1.

Response: Thank you for your careful review, we have added the caption for Figure 1 in the revised version.

Here is the revised version: “Fig 1. The chain mediation model of eHealth literacy and healthcare technology self-efficacy between social support and technophobia.” [Page12, Line 281-282]

3. Please include a caption for table 4.

Response: Thank you for your comment. We sincerely apologize for the oversight of Table 4 in the previous version, which has now been included in full in the revised version. [see revised Page 14, Line 301-308]

[AW

Upper-level Project of the Natural Science Foundation of Shandong Province

grant number: ZR2023MG071

URL: http://kjt.shandong.gov.cn/

the founder gave a financial support in paper submission.

DH

Youth Fund of the Natural Science Foundation of Shandong Province

grant number: ZR2023QG027

URL: http://kjt.shandong.gov.cn/

the founder gave a financial support in paper submission.].

Please state what role the funders took in the study. If the funders had no role, please state: “The funders had no role in study design, data collection and analysis, decision to publish, or preparation of the manuscript.”

Response: Thank you for your comment. We apologize for using the wrong statement, and we have submitted the correct financial disclosure statement in the cover letter.

5. Please include a copy of Table 4 which you refer to in your text on page 11.

Response: Thank you very much for your careful review. We sincerely apologize for the oversight of Table 4 in the previous version, which has now been included in full in the revised version. [see revised Page 14, Line 300-307]

Response: Thank you for your request. We have added 13 references (Refs. 1, 4-6, 32, 34, 37, 39, 40, 41, 43-45) and deleted 2 references (Refs. 38, 42) because of changes to the article. Excluding additions and deletions, we have double-checked the list of references. We have added page numbers to references to make them complete (Refs. 18), and we have changed the names of journals that were not standardized in the references (Refs. 30, 46), as well as changed the format of the author's name in one reference (Refs. 29). We have not cited the retracted references. However, we are open to any further feedback or specific areas that may need further refinement.

Reviewer(s) comments:

The submission is well-written, well-organized, and features an interesting topic. Here are some suggestions to enhance clarity and readability.

Overall response to Reviewer: Thank you for spending time reviewing our manuscript and providing us with a list of constructive comments.

Introduction

The Introduction is comprehensive and well-structured, providing a clear overview of the topic and the rationale for the study. Here are some suggestions to enhance clarity and readability:

1. The opening sentence is crucial in the introduction; the researchers should consider starting with a more engaging sentence to capture the reader’s attention. For example: “coronary heart disease (CHD) remains a leading cause of morbidity and mortality worldwide, particularly among the elderly.”

Response: Thank you very much for your advice. Your suggestion was so helpful that we have changed the first two sentences of the introduction in the revised version.

Here is the revised version: “Coronary heart disease (CHD) remains a leading cause of mortality worldwide, particularly among the elderly[1]. In China, the mortality rate of cardiovascular diseases accounts for the first cause of death, with a prevalence of CHD among individuals aged 60 and older reaching 27.8%[2].” [Page 2, Line 49-51]

2. The submission needs minor revision for smoother transitions between paragraphs to ensure a cohesive flow of ideas.

Response: Thank you very much for this suggestion. We have scrutinized the logic from paragraph to paragraph and added some transition sentences to make the logic flow better in the revised version.

Here is the revised version:

(1) “In this context, technophobia emerges as a key issue.” [ Page 3, Line 81]

(2) “Given its potential to hinder older adults from benefiting from these technologies, identifying the protective factors against technophobia to reduce its impact on this population is essential.” [Page 4, Line 89-91]

(3) “Apart from social support, eHealth literacy is also an essential factor influencing technophobia.” [Page 4, Line 100]

(4) “Additionally, healthcare technology self-efficacy is a key internal protective factor affecting technophobia, playing a critical role in guiding technology use behavior[24]. Defined as an individual’s confidence in using digital healthcare technology, healthcare technology self-efficacy provides a more sensitive measure of an individual's confidence in using digital healthcare technology within a healthcare setting compared to general self-efficacy[25].” [Page 4, Line 107-111]

(5) “Furthermore, eHealth literacy is a protective factor for the self-efficacy of older adults in using health technology[25].” [Page 5, Line 117-119]

3. It is recommended to clarify the existing gap; researchers should succinctly outline the traditional way of accessing medical information and assistance and explain why them are burdensome for healthcare organizations. I mean before or right after this sentence: “Consequently, the traditional way of accessing medical information and assistance burdensome for healthcare organizations.”

Response: Thank you very much for this suggestion. We have explained traditional health care and why it adds to the burden in the revised edition.

Here is the revised version: “With the deepening of aging, the demand for healthcare services among elderly patients with CHD has increased, far exceeding that of other age groups. Traditional healthcare services for CHD are based on outpatient clinics, hospital wards or rehabilitation centers, where healthcare professionals provide disease treatment, medication management, dietary guidance and health education[4]. However, elderly patients with CHD may be reluctant to seek medical care due to the distance from healthcare facilities or the high costs involved, which leads to a higher risk of recurrent events and hospitalizations, ultimately diminishing their chances of survival and quality of life[5]. Additionally, in China, there is an imbalance in the distribution of healthcare resources and a shortage of necessary infrastructure and specialized staff[6]. Consequently, the traditional way of accessing medical information and assistance is burdensome for healthcare organizations.” [Page 3, Line 56-66]

Material and methods

The “Materials and Methods” section is quite comprehensive and well-structured. Here are a few suggestions to improve clarity and readability:

4. The researcher mention that a convenience sampling method was used. It may be useful to briefly discuss the potential limitations of this method, such as selection bias, and how it may affect the generalizability of the findings.

Response: Thank you very much for this suggestion. We have provided a detailed explanation of how the convenience sampling method may lead to selection bias and discussed its potential impact on the results in the study limitations section. Additionally, we also make recommendations for future research in the revised version.

Here is the revised version: “Second, although the participants came from different communities, they were recruited from a single province in China. The convenient sampling method may have introduced selection bias. Because elderly patients with CHD in community outpatient may have milder conditions compared to those in hospital wards, and we excluded individuals in the acute attack period of CHD or with other serious illnesses, this may have led to subtle differences in the results. This limits the generalizability of the results to a broader elderly patient with CHD population. It is recommended that future studies employ probability sampling methods to investigate elderly patients with CHD in various regions and settings.” [Page 18, Line 415-422]

5. The inclusion and exclusion criteria are clearly stated. However, it might be beneficial to justify the exclusion of individuals with acute CHD or other serious diseases, as this could be a significant factor affecting the nuances of your findings.

Response: Thank you very much for your comment. For ethical reasons, we were unable to include patients who are in the acute stage of CHD or suffering from serious illnesses. Therefore, we have added an explanation regarding this limitation in the study limitations section of the study.

Here is the revised version: “Because elderly patients with CHD in community outpatient may have milder conditions compared to those in hospital wards, and we excluded individuals in the acute attack period of CHD or with other serious illnesses, this may have led to subtle differences in the results.” [Page 18, Line 417-420]

6. The list of demographic characteristics is comprehensive. Consider providing a rationale for the choice of these particular variables and how they relate to your study’s aims.

Response: Thank you very much for this suggestion. We selected demographic factors that may affect the outcome variable through a literature review to be analyzed first in univariate analysis and put the statistically significant ones as control variables in the mediation analysis. This approach helps minimize the impact of confounding factors, enabling a clearer understanding of the mediation mechanism and identifying the key factors that play a critical role in the causal pathway. In the revised version, we have explained how demographic factors were selected and supplemented the section on control variables in the statistical analysis.

Here is the revised version:

(1) “Based on a comprehensive review of the literature, we selected demographic factors that may influence the outcome variables and independently developed a questionnaire.” [Page 6, Line 146-147]

(2) “Independent samples t-test or one-way ANOVA was used to compare differences in the demographic characteristics. In the mediation effects analysis that followed, variables that showed significant differences in demographic characteristics were controlled for as covariates.” [Page 8, Line 218-220]

7. The inclusion of Cronbach's alpha values is helpful. It would be useful to provide a short explanation on how these statistics indicate the reliability of the scales used. For example, a Cronbach’s alpha above 0.7 is generally considered acceptable in social science research.

Response: Thank you very much for your suggestion. We have added to the research instrument section how these statistics indicate the scales' reliability in the revised version.

Here is the revised version: “These values are greater than the acceptable value of 0.70, which indicates that the Chinese version of the Technophobia Scale has good reliability.” [Page 6, Line 158-160]

8. Include more details about the development or validation processes of Technophobia and Social Support Measures in the Chinese context if available. This would strengthen the credibility of the scales.

Response: Thank you very much for your suggestion. We have added the reliability of scale validation in the Chinese context to all four scales in the revised version.

Here is the revised version:

(1) “The Chinese version of the Technophobia Scale contains 13 items, 3 dimensions: techno-anxiety, techno-paranoia, and privacy concerns, with responses on a five-point Likert scale from 1 (“strongly disagree”) to 5 (“strongly agree”). The total score ranges from 13 to 65 points, with higher scores representing higher levels of technophobia. The Cronbach’s α coefficient was 0.91 for the total scale and 0.88, 0.83, and 0.75 for the three factors, respectively. These values are greater than the acceptable value of 0.70, which indicates that the Chinese version of the Technophobia Scale has good reliability. The Cronbach’s α of the Technophobia Scale in this study was 0.89.” [Page 6, Line 152-160]

(2) “The SSRS has good reliability and validity with the Cronbach’s α of 0.89 to 0.94.” [Page 6, Line 166-167]

(3) “Cronbach’s α for the Chinese version of eHEALS was 0.91, and the Cronbach’s α for the eHEALS in this study was 0.98.” [Page 7, Line 174-175]

(4) “The Cronbach’s α of the Chinese version of the Healthcare Technology Self-Efficacy Scale was 0.93, the split-half reliability was 0.81, and the re-test reliability after two weeks was 0.89.” [Page 6, Line 183-184]

9. It is good to see ethical approval included. The researcher might consider addressing any specific measures taken to ensure the confidentiality and anonymity of participants.

Response: Thank you very much for your suggestion. In the revised version, we have added specific methods to ensure the confidentiality and anonymity of participants in the data collection section.

Here is the revised version:

(1) “All participants were informed that it was an anonymous survey and they had the right to refuse to participate or withdraw at any time during the study.” [Page 7, Line 192-194]

(1) “All paper questionnaires with the informed consent form were only accessible to the research team to ensure security and confidentiality.” [Page 8, Line 200-202]

10. The process of recruitment and data collection is well described. However, detailing how participants were informed about their rights (e.g., right to withdraw) could strengthen the ethical considerations.

Response: Thank you very much for your suggestion. We have explained that all participants have the right to refuse or withdraw at any time during the study in the revised version.

Here is the revised version: “All participants were informed that it was an anonymous survey and they had the right to refuse to participate or withdraw at any time during the study.” [Page 7, Line 192-194]

11. The researcher mentioned checking completed questionnaires for deficiencies or errors. Elaborating on what specific types of deficiencies or errors were checked could add thoroughness to this section.

Response: Thank you very much for your suggestion. We have added specific types of flaws or errors in the data collection section of the exclusion questionnaire.

Here is the revised version: “The researcher checked the completed questionnaires immediately and asked the participants to provide any missing data. Questionnaires with apparent regularities and logical errors were eliminated, such as a questionnaire with at least a string of more than 10 consecutive identical item responses.” [Page 8, Line 197-200]

Statistical

---

## [Decision Letter · Decision Letter 1]

30 Jan 2025

PONE-D-24-40957R1Social support and technophobia in older patients with coronary heart disease: The mediating roles of eHealth literacy and healthcare technology self-efficacyPLOS ONE

Dear Dr. Wang,

Thank you for submitting your manuscript to PLOS ONE. After careful consideration, we feel that it has merit but does not fully meet PLOS ONE’s publication criteria as it currently stands. Therefore, we invite you to submit a revised version of the manuscript that addresses the points raised during the review process.

We look forward to receiving your revised manuscript.

Kind regards,

Seyedeh Yasamin Parvar, M.D., M.P.H.

Academic Editor

PLOS ONE

Journal Requirements:

Reviewers' comments:

Reviewer's Responses to Questions

**Comments to the Author**

1. If the authors have adequately addressed your comments raised in a previous round of review and you feel that this manuscript is now acceptable for publication, you may indicate that here to bypass the “Comments to the Author” section, enter your conflict of interest statement in the “Confidential to Editor” section, and submit your "Accept" recommendation.

Reviewer #1: All comments have been addressed

Reviewer #2: All comments have been addressed

Reviewer #3: All comments have been addressed

Reviewer #4: (No Response)

Reviewer #5: All comments have been addressed

Reviewer #6: All comments have been addressed

2. Is the manuscript technically sound, and do the data support the conclusions?

Reviewer #1: Yes

Reviewer #2: Yes

Reviewer #3: Yes

Reviewer #4: Yes

Reviewer #5: Yes

Reviewer #6: Yes

3. Has the statistical analysis been performed appropriately and rigorously? 

Reviewer #1: Yes

Reviewer #2: Yes

Reviewer #3: Yes

Reviewer #4: Yes

Reviewer #5: N/A

Reviewer #6: Yes

4. Have the authors made all data underlying the findings in their manuscript fully available?

Reviewer #1: Yes

Reviewer #2: No

Reviewer #3: Yes

Reviewer #4: Yes

Reviewer #5: Yes

Reviewer #6: Yes

5. Is the manuscript presented in an intelligible fashion and written in standard English?

Reviewer #1: Yes

Reviewer #2: Yes

Reviewer #3: Yes

Reviewer #4: Yes

Reviewer #5: Yes

Reviewer #6: Yes

6. Review Comments to the Author

Reviewer #1: The authors have effectively addressed and incorporated the referee's recommendations into the article.

Reviewer #2: The authors have successfully responded to the comments.

Figure 1 quality should be enhaced. The meaning of the asterisks should be added to a figure legend.

Reviewer #3: Dear Editor,

I reviewed the paper titled "Social Support and Technophobia in Older Patients with Coronary Heart Disease: The Mediating Roles of eHealth Literacy and Healthcare Technology Self-Efficacy". This study is a timely and valuable contribution to understanding the relationship between social support and technophobia, with a focus on mediating factors like eHealth literacy and self-efficacy. The authors have tried to address previous reviewer comments, resulting in a well-presented manuscript that is methodologically well-founded.

Here are some minor comments:

Standardize terminology, such as consistently using "older adults" or "elderly patients with CHD" throughout the manuscript.

Avoid redundancy in the discussion and simplify language for improved readability.

Strengthen the articulation of how this study fills gaps in existing research.

Provide more detailed clarification about the communities chosen for the study and their representativeness of the larger population.

Offer more comprehensive suggestions for future study designs, including longitudinal approaches or more diverse sampling strategies.

Emphasize the practical meaning of coefficients and effect sizes beyond their statistical significance. Discuss the actionable significance of findings, such as how social support reduces technophobia by 44.9% through mediating factors.

Provide concrete examples of how social support, eHealth literacy, and self-efficacy can be enhanced in clinical settings.

Include more detailed explanations of statistical terms and findings to aid readers. For instance, expand on the interpretation of mediation effects and bootstrap confidence intervals.

Reviewer #4: The manuscript touches on a topic that is of interest. It is well written and structured. The methodology and statistical analysis employed are appropriate, and the text is well-written. If I may, I would like to humbly offer some suggestions that may help to further enhance its quality.

Abstract

There are two objectives for this article, one is a sentence and the other is a phrase in need of unification and revision.

Main body

Introduction

On page 4, there appears to be an absence of a logical connection between the first and second paragraphs. It would be preferable to refer to social support as one of the supporting factors at the end of the first paragraph.

I would like to suggest that in the introduction, you could perhaps use the following reference to further emphasize the importance of digital health, and specifically telehealth, in the management of CHD: Hayavi-Haghighi MH, Gharibzade A, Choobin N, Ansarifard H. Applications and outcomes of implementing telemedicine for hypertension management in COVID-19 pandemic: A systematic review. PLoS One. 2024 Aug 1;19(8):e0306347. doi: 10.1371/journal.pone.0306347. PMID: 39088489; PMCID: PMC11293715

On page 4, paragraph 2, delete the “in conclusion” from the beginning of the paragraph.

Materials and methods

An inaccuracy was observed in the provision of social support cut points. As with Part technophobia, the provision of complete information regarding this matter, in addition to the utilization of the Likert scoring method, is imperative.

Results

There is no proper caption for Table 4 in the text.

Discussion

The discussion is written in a coherent and argumentative style, but there is a need for greater citation of sources, particularly in the final paragraph of the discussion (p. 16).

Reviewer #5: The manuscript is well-structured and addresses an important topic with clear methodology and analysis. However, I recommend the following for improvement:

Clarity of Mediating Effects: While the mediating roles of eHealth literacy and self-efficacy are statistically supported, further elaboration on the practical implications of these findings would enhance the manuscript’s impact.

Generalizability of Results: The study's sample is limited to a specific province in China. Addressing potential cultural or regional biases and suggesting broader applications would improve the manuscript's robustness.

Language and Grammar: The manuscript's language is generally clear, but minor grammatical refinements would improve readability.

Overall, the manuscript meets academic standards but would benefit from addressing these points to strengthen its quality and broader applicability.

Reviewer #6: I would like to start by saying to the authors that I liked the article very much. Congratulations!

Having carefully reviewed the revised version of the article entitled "Social Support and Technophobia in Older Patients with Coronary Heart Disease: The Mediating Roles of eHealth Literacy and Healthcare Technology Self-Efficacy", I can confidently say that all previous improvements and suggestions have been adequately addressed in this new version. The authors have thoroughly incorporated the addition information that was requested and feedback provided and I think that no further revisions are necessary. The article is comprehensive, well structured, and articulates an important and timely topic that has valuable implications for both healthcare practice and research.

The data presented effectively support the conclusions drawn and the statistical analysis is rigorous, appropriately conducted and clearly communicated. I have no concerns about the methodology or the interpretation of the results. The article is well written and presents complex ideas in a clear and accessible manner, which increases its potential to appeal to a wide audience. Furthermore, the topic is both relevant and important, shedding light on the intersection of social support, technophobia, and the use of eHealth among older patients with coronary heart disease - an area that deserves more attention in contemporary healthcare discussions.

In conclusion, I believe that the article makes a significant contribution to the field and I strongly support its acceptance for publication. The authors have done an excellent job in addressing all concerns and presenting a well-rounded, well-organised manuscript. I have no additional comments or suggestions and am confident that the article will add considerable value to the journal.

7. PLOS authors have the option to publish the peer review history of their article (what does this mean? ). If published, this will include your full peer review and any attached files.

**Do you want your identity to be public for this peer review?** For information about this choice, including consent withdrawal, please see our Privacy Policy .

Reviewer #1: **Yes: ** Azam Shahbodaghi

Reviewer #2: No

Reviewer #3: No

Reviewer #4: **Yes: ** Mohammad Hosein Hayavi-Haghighi

Reviewer #5: **Yes: ** A N M Al Imran

Reviewer #6: No

---

## [Author Response · Author response to Decision Letter 2]

10 Mar 2025

The point-by-point replies to all comments have been provided as follows:

Response: Thank you for your request. We have added 4 references (Refs. 11, 31, 32, 48) because of changes to the article. We have carefully reviewed the reference list and found no errors or citations of retracted articles. However, we are open to any further feedback or specific areas that may need further refinement.

Reviewer(s) comments:

Reviewer #2:

1. Figure 1 quality should be enhaced. The meaning of the asterisks should be added to a figure legend.

Response: Thank you very much for your careful review. We sincerely apologize for this error. We have rechecked Table 1 and found that the p-values were already included, so no additional annotation was necessary. Therefore, we made the following revisions: removed the asterisks superscripted to the p-values and supplemented the missing data in the table. These changes have been completed in the revised manuscript. [Page 10-11, Line 258-259]

Reviewer #3:

1. Standardize terminology, such as consistently using "older adults" or "elderly patients with CHD" throughout the manuscript.

Response: Thank you very much for your suggestion. We have standardized the terminology for the study population. And for consistency with the title, we've changed it to “older patients with CHD” in the revised version.

2. Avoid redundancy in the discussion and simplify language for improved readability.

Response: Thank you very much for your suggestion. We have simplified the language of the discussion section in the revised version by, for example, removing redundant expressions and consolidating sentences with the same meaning.

Here is the revised version:

(1) “The result was consistent with previous research, which observed reduced technophobia in elderly cancer patients with higher levels of family support and social engagement[39].” [Page 15, Line 323-325]

(2) “Due to cognitive and physical decline, older patients with CHD may experience psychological stress when confronted with emerging technologies[40].” [Page 15, Line 325-328]

(3) “Older patients with CHD with robust social support networks can draw upon emotional and practical assistance when learning digital health technologies, thereby reducing fear of technologies, as evidenced by Lee et al.[21]. ” [Page 15, Line 332-335]

(4) “When older patients with CHD feel supported by family, friends, and society, they are better equipped to face the challenges of new technology[42]. Studies have shown that health knowledge seeking and emotional support can all improve eHealth literacy[43], and a high level of eHealth literacy can help reduce technophobia[44].” [Page 15, Line 342-346; Page 16, Line 347-348]

(5) “Encouragement from family and friends can enhance the self-efficacy of older patients with CHD, enabling them to face challenges with greater confidence and resilience, thereby reducing the occurrence of technophobia[27].” [Page 16, Line 356-362]

3. Strengthen the articulation of how this study fills gaps in existing research.

Response: Thank you very much for your comment. We have added an elaboration on how to fill research gaps in the last paragraph of the introduction section.

Here is the revised version: “The relationship between social support, eHealth literacy, and self-efficacy has also been established, particularly in the self-management of older patients with chronic diseases and their use of mobile health technologies[19, 30]. However, to our knowledge, the underlying relationship between these factors and technophobia remains unexplored in existing literature. According to the social ecosystem theory[31], the external environment can influence individual cognition and behavior, which in turn affects the psychological state of older adults when facing technology[27]. ” [Page 5, Line 120-126]

4. Provide more detailed clarification about the communities chosen for the study and their representativeness of the larger population.

Response: Thank you very much for this suggestion. We have added a detailed description of the recruitment area in the Participants section.

Here is the revised version: “Qingdao is an economically developed city in northern China with a population of about 10 million, of which the elderly account for 23.8% of the total population[32]. The four communities were randomly selected from each of Qingdao's four main municipal districts (Shinan, Shibei, Laoshan, Licang), and potential participants lived in both urban and rural communities.” [Page 5, Line139; Page 6, Line 140-143]

5. Offer more comprehensive suggestions for future study designs, including longitudinal approaches or more diverse sampling strategies.

Response: Thank you very much for this suggestion. We have added more specific recommendations in the Study Limitations section.

Here is the revised version:

(1) “Future studies should adopt an intervention or longitudinal approach to examine the real causal relationship.” [Page 18, Line 411-412]

(2) “Future studies should include participants from diverse regions and backgrounds through multi-centre, large-scale study designs to enhance the generalizability of findings.” [Page 18, Line 421-423]

(3) “Future studies should use more objective measures combined with qualitative, observational and experimental approaches to explore the interactions between social support, eHealth literacy, healthcare technology self-efficacy and technophobia.” [Page 18, Line 425-427]

6. Emphasize the practical meaning of coefficients and effect sizes beyond their statistical significance. Discuss the actionable significance of findings, such as how social support reduces technophobia by 44.9% through mediating factors.

Response: We feel great thanks for your professional review work on our article. In this study, the direct effect accounting for 38.5% indicates that even without considering the mediating variables (such as eHealth literacy and healthcare technology self-efficacy), social support alone can significantly reduce technophobia. The larger proportion of the indirect effect compared to the direct effect suggests that social support primarily influences technophobia through mediating variables. Within the indirect effects, the chain mediation effect of eHealth literacy and healthcare technology self-efficacy represents the most significant pathway, contributing nearly half of the total indirect effect. Therefore, in the Discussion section, we explained the implications of these effect proportions and emphasized their relevance in clinical practice.

Here is the revised version:

(1) “Additionally, after adding the mediator variable, the direct effect accounted for 38.5% (-0.257) of the total effect, even without considering the mediator variable, social support can significantly reduce technophobia.” [Page 15, Line 320-323]

(2) “This research found that eHealth literacy and healthcare technology self-efficacy jointly played a chain mediating role in the influence of social support on technophobia among elderly older patients with CHD, with the mediating effect accounting for 44.9% of the total indirect effect (supporting hypothesis 4). This suggests that social support influencing technophobia through the chain-mediated effects of eHealth literacy and healthcare technology self-efficacy is the predominant indirect pathway, contributing nearly half of the total indirect effect.” [Page 16, Line 363-370]

(3) “According to the findings, to alleviate technophobia among elderly older patients with CHD, the relevant departments of hospitals and communities should establish a comprehensive social support system for them. [Page 17, Line 382-385]

(4) “In addition, the findings suggest that the chain-mediated effect of eHealth literacy and healthcare self-efficacy is the core mechanism. When reducing technophobia through social support, efforts should focus on these two aspects.” [Page 17, Line 396-398]

7. Provide concrete examples of how social support, eHealth literacy, and self-efficacy can be enhanced in clinical settings.

Response: Thank you very much for your suggestion. We have added examples of how social support, eHealth literacy, and healthcare technology self-efficacy can be improved in clinical settings in the final paragraph of the discussion section.

Here is the revised version: “According to the findings, to alleviate technophobia among older patients with CHD, the relevant departments of hospitals and communities should establish a comprehensive social support system for them. Healthcare professionals should encourage intergenerational interaction between patients and family members , especially with younger generations, to facilitate digital technology communication[47]. Peer-based technology support groups should be established for older patients with CHD, facilitating experience sharing through both offline activities and online communities[48]. Community healthcare centers can organize training sessions on technological skills, providing patients with spaces for learning and interaction[48]. In addition, the findings suggest that the chain-mediated effect of eHealth literacy and healthcare self-efficacy is the core mechanism. When reducing technophobia through social support, efforts should focus on these two aspects. Family members and professionals should prioritize improving patients’ basic technical competence and critical thinking skills to identify misinformation[21]. Instructors should provide immediate encouragement after patients master each technical skill to enhance self-efficacy. For anxiety management, instructors can guide patients to adopt positive self-affirmations during technology use.” [Page 17, Line 382-402]

8. Include more detailed explanations of statistical terms and findings to aid readers. For instance, expand on the interpretation of mediation effects and bootstrap confidence intervals.

Response: Thank you very much for your advice. We have added explanations of chained mediation effect and the 95% confidence interval in the revised version.

Here is the revised version:

(1) “Secondly, the chain mediation effect was tested using Model 6 from the SPSS-PROCESS macro program, which refers to the indirect effect in a causal pathway where the influence of an independent variable (X) on a dependent variable (Y) is transmitted sequentially through multiple mediators (e.g., M₁, M₂) in a specified order.” [Page 9, Line 226-230]

(2) “The 95% CI is a statistical range used to estimate the plausible values of mediation effect. If the study were repeated 100 times, approximately 95 of the calculated intervals would contain the true parameter value.” [Page 9, Line 233-235]

Reviewer #4:

1. There are two objectives for this article, one is a sentence and the other is a phrase in need of unification and revision.

Response: Thank you very much for your advice. Your suggestion was so helpful that we have changed the first sentences of the abstract in the revised version.

Here is the revised version: “The purpose of this study was to explore the relationship between social support, eHealth literacy, healthcare technology self-efficacy, and technophobia. It also analyzed the mediating effect of eHealth literacy and healthcare technology self-efficacy between social support and technophobia.” [Page 2, Line 28-30]

2. On page 4, there appears to be an absence of a logical connection between the first and second paragraphs. It would be preferable to refer to social support as one of the supporting factors at the end of the first paragraph.

Response: Thank you very much for your suggestion. Based on your guidance, we introduced social support as one of the supporting factors at the end of the third paragraph of the introduction. And we also modified the first sentence of the fourth paragraph to make it articulate.

Here is the revised version:

(1) “As an important coping resource, social support is considered a key external protective factor[18, 19].” [Page 4, Line 91-92]

(2) “Social support has been found to influence technology adoption and utilization among individuals, and there is a significant negative correlation between social support and technophobia in older adults[20].” [Page 4, Line 93-95]

3. I would like to suggest that in the introduction, you could perhaps use the following reference to further emphasize the importance of digital health, and specifically telehealth, in the management of CHD: Hayavi-Haghighi MH, Gharibzade A, Choobin N, Ansarifard H. Applications and outcomes of implementing telemedicine for hypertension management in COVID-19 pandemic: A systematic review. PLoS One. 2024 Aug 1;19(8):e0306347. doi: 10.1371/journal.pone.0306347. PMID: 39088489; PMCID: PMC11293715

Response: Thank you very much for your suggestion, it is very helpful. We have incorporated the perspective from this literature into the Introduction section and included a citation to the source in the revised version.

Here is the revised version: “Most patients and healthcare professionals believe that digital health technologies can provide convenient and effective medical services[11].” [Page 3, Line 74-75]

4. On page 4, paragraph 2, delete the “in conclusion” from the beginning of the paragraph.

Response: Thank you very much for this suggestion. We have deleted “in conclusion” in the revised version.

Here is the revised version: “Previous studies have shown that social support, eHealth literacy, and healthcare technology self-efficacy play an important role in influencing technophobia in older patients with CHD.” [Page 5, Line 118-120]

5. An inaccuracy was observed in the provision of social support cut points. As with Part technophobia, the provision of complete information regarding this matter, in addition to the utilization of the Likert scoring method, is imperative.

Response: Thank you very much for this suggestion. We have supplemented the scoring of the Social Support Rating Scale and corrected errors in the cut-off points.

Here is the revised version: “The Social Support Rating Scale (SSRS) which was compiled by Xiao in 1994[34], was used to measure social support. The scale includes 10 items and 3 dimensions: subjective support, objective support, and utilization of support. Items 1-4 & 8-10: Select one option per item (1-4 points). Item 5: A-D options (4-point scale: 1=none to 4=full support). Items 6-7: 0 points without sources; score = number of sources listed. The total score on the scale ranges from 12 to 66, with higher scores representing more social support. A total score of 12–22 indicates a low level of social support, 23–44 indicates a medium level of social support, and 45–66 indicates a high level of social support. The SSRS has good reliability and validity with Cronbach’s α of 0.89 to 0.94. The Cronbach’s α of this scale in this study was 0.87.” [Page 6, Line 166-167; Page 7, Line 168-174]

6. There is no proper caption for Table 4 in the text.

Response: Thank you very much for your suggestion. We have modified the title of Table 4.

Here is the revised version: “Table 4. The mediating effect of eHealth literacy and healthcare technology self-efficacy between social support and technophobia.” [Page 14, Line 307-309]

7. The discussion is written in a coherent and argumentative style, but there is a need for greater citation of sources, particularly in the final paragraph of the discussion (p. 16).

Response: We feel great thanks for your professional review work on our article. We have revised the final paragraph of the Discussion section based on other suggestions and added the relevant references as per your guidance.

Here is the revised version: “According to the findings

---

## [Decision Letter · Decision Letter 2]

8 May 2025

Social support and technophobia in older patients with coronary heart disease: The mediating roles of eHealth literacy and healthcare technology self-efficacy

PONE-D-24-40957R2

Dear Dr. Wang,

We’re pleased to inform you that your manuscript has been judged scientifically suitable for publication and will be formally accepted for publication once it meets all outstanding technical requirements.

Kind regards,

Seyedeh Yasamin Parvar, M.D., M.P.H.

Academic Editor

PLOS ONE

Additional Editor Comments (optional):

Reviewers' comments:

Reviewer's Responses to Questions

**Comments to the Author**

1. If the authors have adequately addressed your comments raised in a previous round of review and you feel that this manuscript is now acceptable for publication, you may indicate that here to bypass the “Comments to the Author” section, enter your conflict of interest statement in the “Confidential to Editor” section, and submit your "Accept" recommendation.

Reviewer #4: All comments have been addressed

2. Is the manuscript technically sound, and do the data support the conclusions?

Reviewer #4: Yes

3. Has the statistical analysis been performed appropriately and rigorously? 

Reviewer #4: I Don't Know

4. Have the authors made all data underlying the findings in their manuscript fully available?

Reviewer #4: Yes

5. Is the manuscript presented in an intelligible fashion and written in standard English?

Reviewer #4: Yes

6. Review Comments to the Author

Reviewer #4: (No Response)

7. PLOS authors have the option to publish the peer review history of their article (what does this mean? ). If published, this will include your full peer review and any attached files.

**Do you want your identity to be public for this peer review?** For information about this choice, including consent withdrawal, please see our Privacy Policy .

Reviewer #4: **Yes: ** Mohammad Hosein Hayavi-Haghighi

---

## [Editor Report · Acceptance letter]

PONE-D-24-40957R2

PLOS ONE

Dear Dr. Wang,

I'm pleased to inform you that your manuscript has been deemed suitable for publication in PLOS ONE. Congratulations! Your manuscript is now being handed over to our production team.

Kind regards,

on behalf of

Dr. Seyedeh Yasamin Parvar

Academic Editor

PLOS ONE